# Norwegian Physical Education Teacher Education Students' Perceptions of the Subject Physical Education: A Qualitative Study of Students' Reflections before Starting Their Studies

Ove Østerlie *[ID] and Geir Olav Kristensen

Department of Teacher Education, Faculty of Social and Educational Science, NTNU—Norwegian University of Science and Technology, 7034 Trondheim, Norway
* Correspondence: ove.osterlie@ntnu.no

**Abstract:** Studies show that students' view of the subject of physical education (PE) is often dominated by sports discourse, and that the profession of higher education often fails to balance this view, which, in many contexts, does not align with the aims of governing documents in PE. The purpose of this study was to gain insight into how future students in physical education teacher education (PETE) perceive the subject. Written responses from 112 students at the start of their PETE study were analysed within the framework of reflexive thematic analysis. Based on questions about activity habits, we found that most students were active individuals who engaged in traditional physical activities such as ball games and basic training in their leisure time. Analysis of the students' answers generated four themes: (1) Activity and bodily learning, (2) Motivation and joy of movement, (3) Health for life, and (4) "Bildung". We argue through these four themes that the students' perception of the subject with a focus on much physical activity through a variety of activities—physical learning, motivation, and joy of movement—are in line with governing documents and physical literacy. It is also clear that health discourse is strongly represented in the students' perception of the subject and that physical education is an important contributor in the students' formation process. However, some aspects of the curriculum seemed to be of lesser interest to the students. In conclusion, the students' perceptions and experiences can serve as a starting point for change, learning, and development in physical education teacher education.

**Keywords:** physical education (PE); physical education teacher education (PETE); bodily learning; motivation; health

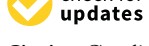



## 1. Introduction

For the mandate and goals in the curriculum in physical education (PE) to be fulfilled to the greatest extent possible, the school depends on well-qualified teachers to teach PE. Ward and Kim [1] claim that the development of the PE teacher's pedagogical and didactic knowledge plays a major role in the pupils' learning outcomes. The main goal of educating teachers is that the education should have a lasting and strong effect on the exercise of their future professional life, but there is uncertainty about the extent to which this goal is achieved [2]. The education of teachers in PE has received more attention in recent decades, also through highlighting the challenges that this type of education faces [3]. It has previously been pointed out that research into the motivation to become a PE teacher is lacking [4]. Moreover, little knowledge has been produced in this area in recent years. Nevertheless, there is some prior research that this study can draw upon. Interest in sport and physical activity turns out to be the main motive for many who want to start studies to become PE teachers [5,6]. In Sweden, Romar and Åström [7] found that students starting PETE had largely positive experiences as students in the subject and that they had a great interest in sports and physical activity. These findings reflect results from studies in England [8]. González-Calvo and Gerdin [9] found that Spanish students who

wanted to become PE teachers were highly influenced in their choice of education through their previous experiences as students in the subject. Those who enjoyed PE as students, a subject which, in Spain, is significantly influenced by sport [10,11], would become teachers in the subject to re-create and pass on the positive climate they themselves had experienced. The researchers behind the study then assumed that the students would in the future reproduce a sports paradigm without much reflection on either the negative or positive aspects of the subject. In the same study, they observed that those who had chosen to study to become PE teachers, despite negative experiences in the subject as students, tended to want to focus more on health discourse as future teachers and to create a more inclusive and caring learning climate [9].

Several studies show that physical education teacher education (PETE) neither changes the students' view of the subject to a large extent [12] nor has a major impact on the students as future PE teachers [13]. In contrast, PETE seems to have a good influence on students in those contexts where the education is well constructed and in close contact with the field of practice [14–17]. Furthermore, Standal and Moen [15] and Moen and Green [18] add that PETE often becomes an arena for the reproduction of students' preconceptions of PE as a sports-dominated subject, where both teachers and students focus mostly on instruction in sports skills. This reflects an international trend in PE [3,17,19,20], which also shows that experiences as a (pre-university) student seem stronger than formal education when it comes to which perspectives PE teachers have on the subject [4]. Later research points to the importance of understanding the students in different ways, including their motivation for change, when they enter PETE in order to reverse this trend [6,21]. In this sense, it is essential to understand how individuals who will be teaching the subject both understand, interpret, and act based on the governing documents in PE. An important perspective in this context is understanding which individuals go into PETE, why they want this career, and what they think about the subject. The desired student outcome in Norwegian PETE is closely related to the national curriculum in PE [22]. While not specifically mentioned in this curriculum, physical literacy (PL) is argued to be a core outcome of PE in several contexts [23], alongside positively impacting leisure physical activity and mental health in youth [24]. Hence, PL and the national curriculum in PE are used as a conceptual framework in this study to understand students' perceptions of PE.

## 2. The Norwegian Curriculum in PE and Physical Literacy

In this study, we have chosen to view our findings through a theoretical foundation based on the curriculum in PE and PL. Considering the age of the sample in the study, we can assume that the participants themselves have been pupils in PE when the current curriculum was the "Curriculum for Knowledge Promotion 2006" (LK06, revised in 2015) [25]. Since we want to observe our findings in light of the curriculum which these students, in turn, will be working towards in school, we will also include the Curriculum for Knowledge Promotion 2020 (LK20) [22] in this background.

Through three core elements, movement and bodily learning, participation and interaction in movement activities, and outdoor activities and nature wandering, the subject will now facilitate a wide selection of different movement activities. Students must explore their own identity and self-image to a greater extent and understand the connections between movement, body, exercise and health [22]. LK20 is a continuation of LK06, which described a purpose of the subject which was linked to activity and training based on personal background and ability. LK06 also focused on the creation of meaning in connection with movement activities, and the subject is important for practicing movement activities in the broadest sense. The teaching should also promote positive self-understanding and identity with the body. The student should experience joy in movement, inspiration, and self-understanding through bodily movement, outdoor life, and joint activities with others, and in the last part of upper secondary school teaching, emphasis should be placed on positive movement experiences and a good bodily self-concept in order to promote a lasting, physically active lifestyle [25]. Several researchers have suggested that both students and teachers have a relatively narrow

view of the purpose of the subject, where physical activity acts as a break in the school day, i.e., without any focus on learning, is most important [26,27]. The observed view of students and teachers on health in physical education also seems to be narrowed down to dealing with physical fitness as prevention of disease [28,29].

Internationally, the concept of PL is gaining greater prominence in physical education [30,31]. Historically, PL has been interpreted with, and attributed to, different characteristics and content within different academic communities, and attempts to establish a global definition have been unsuccessful [32]. Recent research has interpreted these different directions or traditions differently and recommends that we must move away from thinking of one definition, instead acknowledging that the concept must be interpreted in context, and that different definitions can co-exist [30]. PL is not something that a student in PE can learn; rather, it is an expression of a number of factors that are essential skills in the subject. One can well compare the concept of PL with the concept of "the weather". Weather is a construct—it is not itself a real thing; several elements, each operationalisable, and thus a real thing, contribute to weather, e.g., temperature, humidity, wind speed, barometric pressure, etc. [33]. PL is therefore not a term that is used explicitly in the PE curriculum in Norway, but the operationalised elements that the term consists of can be seen through the three core elements described earlier (See also Table 1). By understanding the subject of PE not only through the curriculum but also in terms of how the concept or philosophy of PL appears in the curriculum, we can gain a better understanding of the students' perceptions of the subject.

In this study, we understand PL through what Young and Alfrey [30] call a different cosmos, where a cosmos represents a tradition or direction within interpretations of the concept of PL. The first cosmos is *PL as health-promoting physical activity*. This way of understanding PL has dominated globally and is visible in PE through the subject's mandate and political legitimation as an arena for physical activity with the aim of preventing ill health and inactivity. For example, LK20 states in connection with the subject's relevance that "The subject should motivate students to maintain a physically active and health-promoting lifestyle after finishing school and in future working life" [22]. This cosmos is also consolidated in LK20 through the interdisciplinary topic of "health and life skills". The second cosmos is *PL as motor competence* with attention on the development of basic movement skills. This way of understanding PL is widespread in PE and comes to light through, for example, the core element, "movement and bodily learning", with the text, "bodily learning refers to developing all-round motor-skills and awareness of the body, and stimulating the joy of movement". [22]. In a Norwegian context, PL has been translated and understood as "movement literacy" [34], "physical-motor skills" [35], or "motor-bodily ability" [36]. This draws a clear connection to this second cosmos. The third cosmos is *PL as phenomenological embodiment*. This more holistic interpretation of the term was, among other things, a response to PE moving in the direction of elitism with a strong focus on performance and the Cartesian view of head–body dualism [37]. Through an embodiment perspective, researchers point out that experiences in one area of knowledge can have significance for another area of knowledge; for example, from bodily processes to social or cognitive learning [38]. This broad understanding of the term has resulted in a definition stating that "Physical literacy can be described as the motivation, confidence, physical competence, knowledge and understanding to value and take responsibility for engagement in physical activities for life" [39]. An attempt has been made at a conceptual validation of these six components, where it was argued that knowledge and understanding did not fit as well into the model as the remaining components [40]. In the Norwegian context, operationalising this direction through the concepts of physicality, tact, and physical competence has been attempted [41]. Physical literacy as phenomenological embodiment has, in recent decades, influenced the development of governing documents in physical education in several countries, e.g., Wales [42], and can be said to be the cosmos that can be recognised to the greatest extent in LK20 by comparing the subject's relevance and central values" and IPLA's definition. It is also only this cosmos that is recognised in all three core elements in LK20, as shown in Table 1.

**Table 1.** Overview of the final themes that were generated in the analysis, as well as all tentative themes with examples of student quotes for each of them, and how the core elements of LK20 and the various PL cosmos are positioned in relation to the final theme.

| Final Theme | Tentative Theme | Example Student Quotes | Core Element in LK20 | PL Cosmos |
|---|---|---|---|---|
| Activity and bodily learning | activity | *I think that physical education is a very important part of primary school, both because it is important to make pupils more physically active and to introduce them to different forms of physical activity* (female student). | Movement and bodily learning | PL as health-promoting physical activity PL as motor competence PL as phenomenological embodiment |
| | learning | *It is a subject where students who may have problems with purely theoretical subjects can develop themselves, and most of the theoretical content in this subject can be "visualised". (e.g., nutrition with food, and learning about the body through actively using it)* (female student). | | |
| | teachers | *Lots of sports. Teacher with poor competence who is not trained in the subject. Too little activity in the lessons themselves. The boys who are good at football slip through. Effort should be given a greater place. Assessment should be seen in the light of progression with regard to the pupils' starting point* (male student). | | |
| Motivation and joy of movement | motivation | *For me, physical education should include, engage, and motivate students for further exploration. It should be an arena where one discovers new activities* (male student). | Movement and bodily learning Participation and interaction in movement activities | PL as health-promoting physical activity PL as motor competence PL as phenomenological embodiment |
| | mastery | *Physical education can give students versatile movement experiences and experience mastery, which in turn can inspire a more physically active lifestyle* (female student). | | |
| | variation | *I perceive physical education as a varied and fun subject in primary school. I think physical education is an important subject in order to give pupils varied movement challenges in different environments* (female student). | | |
| | play | *I understand that PE in primary school should be characterised by a lot of play and general activity. I think it can make a positive contribution both in terms of their social characteristics, as well as providing them with a healthy and good lifestyle with enjoyment of movement* (male student). | | |
| | different | *I see physical education in primary school as an arena where pupils can express themselves and learn in a different way than in the more traditional school subject. Here, they get to express themselves physically, and to a large extent through interaction with others* (female student). | | |
| | testing | *Physical education in primary school was not a positive experience for me. I think the classes were fun because there was no theory. But there were often arguments, a lot of focus on the good ones, not fair play (especially among boys), there were a lot of tests that gave grades based on a time, for example, and it's a subject where it's incredibly easy to compare yourself to others in... There must be less focus on comparison* (female student). | | |

Table 1. *Cont.*

| Final Theme | Tentative Theme | Example Student Quotes | Core Element in LK20 | PL Cosmos |
|---|---|---|---|---|
| Health for life | diet | *The subject could have had more focus on nutrition, so that the pupils from a young age have the competence to improve their diet and see the benefits of it when the pupils grow up* (male student). | Movement and bodily learning | PL as health-promoting physical activity PL as phenomenological embodiment |
| | Importance of physical activity | *I think physical education in primary school is very important; many young people today do very little physical activity. That's why I think it's great that you do it a bit through school. But also, for those pupils who are in a lot of physical activity outside of school* (male student). | | |
| Bildung | cooperation | *The positive thing is that the pupils are trained in cooperation and incorporate fair play* (male student). | Participation and interaction in movement activities | PL as phenomenological embodiment |

Notes: LK20 = Curriculum for Knowledge Promotion 2020 (LK20) [22]; PL = physical literacy.

There is thus some knowledge about how education shapes and socialises students into the teaching role, but we know little about what perception the students have of the subject before they start the education, at least in the Norwegian context. The present study seeks to answer the research question: How do future Norwegian PETE students perceive the school subject PE? We have used a conceptual framework consisting of the phenomenon of PL, the Norwegian national curriculum in PE, and current knowledge in the field to understand the students' written, self-reported perceptions of school PE.

**3. Methods**

The following section elaborates on the study participants, the survey used to generate data, ethical considerations, and an amplification on the analysis of the data including considerations of the study trustworthiness.

*3.1. Participants*

In this qualitative study, 180 students who had chosen the subject physical education 1 as part of their primary and lower secondary teacher education were invited to participate by "purposeful sampling" [43]. Physical education 1 is a subject of 30 credits that can be chosen at different times during the course of teacher education. All the participating students had chosen and been given a place on the course, but the data generation took place before teaching in the course started. Of those invited, there were N = 112 (response rate 62.2%) students who submitted their answers electronically via the Surveymonkey service or the distributed questionnaires (first year of study: 53; third year of study: 8; foirth year of study: 51). There were 62 women and 50 men, with an average age of 22.4 years (SD = 2.0), who finally gave their consent to participate by submitting their answers.

*3.2. Survey*

The students who wanted to participate in this study filled in a questionnaire in which they were asked to provide demographic information such as gender, age, and activity habits. To generate data about the students' activity habits, the following questions were asked: "Do you engage in physical activity? What kind?" To generate data for the research question, the students were asked to write an answer of up to 500 words to the following question: "How do you perceive physical education in primary school?" Using open-ended questions, we invited completely unstructured responses: narratives unconstrained by a fixed set of possible responses [44]. The data were generated in January (92 student responses) and August/September (20 student responses) 2019 based on the semester in which the students had chosen the subject. All the students belonged to one university in Norway.

*3.3. Ethical Considerations*

The students who were invited to participate in this study were thoroughly informed about the study's content, objectives, and expected publications. The study was reported to the Norwegian Agency for Shared Services in Education and Research (NSD) (#320518), and their assessment was that no direct or indirect information that could identify individuals in this study should be processed. Since no names or IP addresses were registered for those who submitted electronically, no answers could later be withdrawn. In accordance with NSD's guidelines, it was pointed out that anonymity would be assured, and that no participant could be identified via either the data or publications based on the data. The electronic service used was assessed by NSD as being secure in terms of anonymity. Furthermore, the generated data were stored on a secure server at the authors' institution in accordance with internal routines.

*3.4. Analysis*

The questions about activity habits were analysed on an overall level to obtain an overview. These data were not used to compare groups such as gender or participation in

individual/team sports. To answer the research question in this study, the generated data on the students' perception of the subject were analysed by reflexive thematic analysis (TA) [45]. This choice was made in view of the researchers' position in the social constructivist tradition and the recognition that much knowledge about PE and PETE had to be reflected upon in the process of understanding and interpreting the data generated in the study.

The process of coding and generating themes was guided by reflexive thematic analysis [45] and involved six steps, in which both organic and structured reflection [46] characterised the analysis. First, the data were read through by both authors to familiarise themselves thoroughly with the content. In the second step, both researchers carried out an initial parallel analysis, where the answers were condensed into codes. In this phase, the researchers placed particular emphasis on adopting an inductive approach, where the researcher starts the analytical process from the data and works "bottom-up" to identify meaning without importing ideas. In practice, any researcher will have to approach the data with preconceived ideas based on their existing knowledge and views. Coding inductively does not therefore mean that the researcher is in a "blank state"; instead, it means that the starting point for the analysis is in the data, instead of in existing concepts or theories [47]. In the third step, the codes were collected and generated into tentative themes in a process where we as researchers were active and creative in producing these, i.e., doing more than observing (tentative) themes that emerged from the data. This has recently been pointed out by Braun and Clarke [48] as important in authentic (reflexive) TA. This part of the analysis was also conducted in parallel by both researchers, followed by a comparison and discussion of the tentative themes. This approach is claimed by Kvale [49] to strengthen what he called communicative validity through the researchers' conflicting knowledge claims, since themes that emerge from the data are constructions of the researcher [50]. This action is also referred to as researcher triangulation and is considered an important contribution to credibility in general in qualitative studies [51] and also in studies that specifically use thematic analysis [52]. In this phase, the thematic map technique [45] was used in order to see connections between codes, tentative themes, and any sub-themes. At the same time, both researchers also noted down reflexive content from the answers, which would later provide content for the discussion of the various themes. In the fourth step, the tentative themes were revised and further developed, where the researchers together assessed the adaptation of themes and any sub-themes, with a reflexive look back at the answers and codes, in order to see whether the condensation of meaning had the closeness to the data material needed to create as much credibility as possible in the analysis [52]. In the fifth step, which focused on defining and naming the final themes, the researchers sought to ensure that the themes and theme names clearly, comprehensively, and concisely, captured what was meaningful about the data with respect to the research question. Table 1 shows an overview of the final themes that were generated in the analysis, as well as all the tentative themes, together with student quotes that illuminate and exemplify the themes, and how the core elements of LK20 and the various PL cosmos are positioned in relation to the final theme.

The sixth and final phase is to present the results. In line with recommendations in reflexive thematic analysis [45], the findings and discussion are combined since the analysis was very inductive and the findings are elucidated through a conceptual theoretical basis. The scholarly process of making connections to existing research and literature on the topic of interest and weaving this into the written results and discussion can offer some final moments of inspiration and a deeper insight into the analysis. Therefore, Braun and Clarke [53] urge researchers to see this phase as the final phase of the analysis rather than a mere writing exercise.

In addition to the reflections on credibility described earlier, the authors have emphasised transparency in the presentation of all the choices made in the study. In qualitative studies, transferability or generalisability are not necessarily the most important concepts for quality, but a good "craft" [54] and offering transparency in the process so that the reader can decide to what extent the presented findings and results are valid or credible

for their own context [55,56]. In the next section, selected parts of the students' responses will be included in the text to describe the themes more thoroughly and to strengthen the credibility of the study. Through what Dahler-Larsen [57] refers to as "authenticity", the reader should be able to see a clear connection and chain of evidence from the data to the conclusion when reporting qualitative data.

## 4. Findings and Discussion

Initially, we provide a brief presentation of the findings regarding the students' activity habits. Out of 112 students, 53 engaged in strength training, 45 played soccer, 40 engaged in endurance training, 21 answered that they went to the gym, and 8 engaged in basic training, without specifying whether it was for strength or endurance training. This gives us a picture that those who enter PETE are both fond of being physically active and that they are still active in traditional sports at a relatively high age. We do not see that this group of students is largely active in more unconventional or modern activities. If we compare the findings on activity habits with national surveys on the content of physical education in Norway, we see a correlation where the content of physical education is largely ball games and basic training [58]. Furthermore, this picture may coincide with the findings of Curtner-Smith [5], González-Calvo and Gerdin [9] and Romar and Åström [7], which showed that interest in sport and physical activity was the main motivator for many in the choice to start studies to become PE teachers. The question is whether these interests that the students have will also make them prioritise this type of activity in their future teaching, as suggested in the study by González-Calvo and Gerdin [9]. Standal and Moen [15] and Moen and Green [18] have found that PETE often becomes an arena for the reproduction of the students' prior understanding of PE as a sport-dominated subject, where both teachers and students primarily focus on instructing athletic skills.

### 4.1. How Do PETE Students Perceive the Subject PE?

As a result of the analysis of the students' responses to how they perceive the subject, 13 tentative themes were created, which became the basis for 4 final themes. Table 1 provides an overview of these themes, as well as examples of student responses that illuminate and exemplify each individual tentative theme. Furthermore, we will discuss what connections the students' perceptions have with LK20 and PL.

#### 4.1.1. Activity and Bodily Learning

A theme from the analysis was the importance of motor learning and development. The students pointed out that pupils must experience and learn several different movement activities, including non-traditional activities, so that the experience basis is large. "I think that physical education is a very important part of primary school, both because it is important to make pupils more physically active and to introduce them to different forms of physical activity" (female student). Our interpretation of this is that the students' perception of the subject is largely in line with the intentions of LK20, which emphasises various movement activities, play, and practice to a greater extent than previously. This is somewhat in contrast to previous findings, which suggest that PETE students who are also active in sports, as they are in this study, may have a tendency to emphasise and value a sports discourse in the subject [9]. On the other hand, with our data, we cannot say anything about how the students we have included in the study will be influenced by their view of the subject in their future working life. Nevertheless, we can advance the idea that today's students in PETE, many of whom are active athletes, enter this education with a view of the subject that is more in line with the subject's mandate than previously observed in the Norwegian context [13,15]. Our interpretation of the students' perceptions must also be seen in light of the fact that parts of the research we compare with have been performed in a more specialised or sport education context, where the student body is possibly to a greater extent characterised by a sports discourse. Regarding PL, this theme consolidates itself within all three cosmoses described at the beginning and in Table 1, but especially

within PL as motor competence. Understood as motor competence, PL has infiltrated both in LK20 and in the students' perception of the subject, where they believe that most of the time must be spent on physical activity and basic motor competence. This can also be recognised internationally, where analysis tools have been developed in some countries to map basic motor competence in students in physical education [59], and where basic motor competence is a concrete goal in the curricula for the subject [60]. Internationally, we also see that PL is often considered to be equal to physical competence [61] or basic motor skills [62].

Although the students' perceptions of the subject are understood to be in line with the subject's overall goals and mandate and the concept of PL, there were several areas of LK20 that were not mentioned by the students, such as, e.g., outdoor life, dancing, swimming, and lifesaving. This may have a connection with the fact that in PE, they seem to be low-priority areas [58]. If the students do not associate PE with these areas, it may be necessary to shed extra light on this in PETE. The students seem to have the opinion that movement and activity are more important than the more technical exercises you encounter in sports. Many students mention that the teachers should make demands on the students when it comes to learning, and that the teachers must focus on PE as a learning subject by setting learning goals for the lessons. These thoughts about the subject emerge through statements that the subject must provide knowledge about how the body reacts when carrying out various activities, but it is pointed out by the students that learning must take place through bodily movement. "I see physical education as an active subject where you can learn a lot about activity, health and movement and the development of learning and movement" (female student). This view of PE, where the students believe that pupils must be able to experience being in physical activity at the same time as being learning individuals, is a somewhat surprising finding considering that until now, they have only been pupils in the subject and have no formal education as PE teachers.

Furthermore, the students indicate that this requires good planning from the teacher, and that good planning requires the teacher to have good competence. A point that was prominent in the analysis was the students' preoccupation with the teacher's competence, and that little or no competence (in the form of formal education as a teacher in the subject) was negative, both in terms of content and the teacher's ability to motivate all the pupils. "With good teachers, it can lead to further interest in a good lifestyle and maintaining good health" (male student). This opinion is in line with research that shows that "teacher quality", in the form of trained teachers, is one of the most important criteria for good learning in physical education [1].

> "Lots of sports. Teacher with poor competence who is not trained in the subject. Too little activity in the lessons. The boys who are good at football slip through. Effort should be given a greater place. Assessment should be seen in the light of progression regarding the student's starting point". (male student)

That PETE students see this as a challenge is uplifting, as they value the subject such that they believe the teacher needs to have formal competence. Before even starting PETE, this perception can stem from their own experiences where they had teachers lacking formal competence in the subject, resulting in poor learning. It can also be influenced by the socialisation process, where they have learned that the subject matter, the teacher's expertise, and the value of learning are highly regarded by those around them. This can be contrasted with research that suggests that both students (and teachers) do not value PE beyond the value of being active [26], and that the subject largely lacks a focus on learning [27]. The students' statements in favour of more activity in PE were often based on the wish that the pupils find an activity they want to do in their leisure time, or that the activity increases motivation and desire for learning in general, which brings us to the next theme.

4.1.2. Motivation and Joy of Movement

The students express that the subject should always aim to promote enjoyment of movement by motivating physical activity.

> "I perceive PE as a different subject that motivates pupils who struggle with more theoretical subjects. If the teachers vary their teaching approach, this subject can reach more pupils, and by focusing on different skills it provides a sense of achievement by learning something new and being able to do something that was previously learned. They can learn and enjoy something that will be very important later in life. The enjoyment of movement is a key aspect". (female student)

Based on the extent to which these aspects were mentioned in the answers, we understood this as the perceived main purpose of the subject, together with activity, and the students see that these two themes also have a strong connection. We draw this conclusion after reading statements where these terms are used together, such as in this statement: "For me, physical education should include, engage, and motivate students for further exploration. It should be an arena where you discover new activities" (male student). If we see this in the context of the curriculum in the subject, we see that much coincides with the subject's relevance, about which it is stated, among other things, that "Physical education is an important subject for stimulating lifelong joy of movement and a physically active lifestyle based on personal qualities and abilities" [22].

The students perceive that mastery is an important part of PE and that pupils must know that they can achieve this, but they must also be stimulated to push their own limits. It is therefore important, according to the students, to make adjustments so that all pupils are given challenges at their level. "PE in primary school is an important subject and many pupils' favourite subject. I think PE should be organised more so that all pupils feel they have mastered the subject" (female student). These perceptions come to light in the analysis through descriptions of the subject as varied, different, and as an arena for trying out non-traditional activities. Competition is also mentioned by the students as an important element in the subject, but only if it is "used correctly", because it can help increase motivation and the level of activity. The students emphasise that PE must be delivered in such a way that the pupils can challenge themselves. In the encounter with new activities and sports, an interest is created that can contribute to pupils starting with the activity/sport in their leisure time. According to the students, a playful approach to the subject can contribute to motivation and lifelong enjoyment of movement for all pupils, even those who are not active in their free time. Students also focus on development rather than results and sports. "The development focus must be promoted more and the joy/self-worth in the subject must be central" (female student). The students mention that comparison and testing must be toned down. Variation in methods and content is important, according to the students, so that the pupils have the opportunity to experience the many different sides of physical activity, which is important for the pupils' well-being, learning, and motivation. In the light of PL as phenomenological embodiment, we recognise the students' perceptions of PE as a subject in which a holistic way of thinking should embrace both the diversity of the pupils and their different competences and prerequisites. This is in line with the origin of this way of understanding the term, which was a response to elitism and the performance focus that had developed in PE [37].

Our findings are somewhat contradictory to previous research, where Standal and Moen [15] and Moen and Green [18] argue that PETE often becomes an arena for the reproduction of the students' pre-understanding of PE as a sports-dominated subject, where both teachers and students mostly focus on the instruction of sports skills. We do not find evidence in our analysis of the student responses to say that the students, even though they are active sportsmen themselves to a large extent, have a perception of the subject as sports dominant. We see in this study that the students largely have a view of

the subject that is in accordance with the subject's mandate and goals in schools, where motivation for physical activity that will create enjoyment of movement dominates.

### 4.1.3. Health for Life

The students perceived that the content of PE should also be about health and, preferably, knowledge about health and nutrition, if such is possible in an interdisciplinary perspective. One student wrote: "The subject could have focused more on nutrition, so that pupils from a young age have the skills to improve their diet and see the benefits of that when they grow up" (male student). Several students specifically mentioned nutrition, without this being directly mentioned in LK20. Based on the student responses, we interpret that health discourse is prominent in the students' perception of PE. They perceive the subject as a good arena for promoting a healthy lifestyle, and for promoting good health through a lot of physical activity. This is somewhat in contrast to a perspective on health as a starting point for learning, which researchers have pointed out as a possibly more positive angle on health discourse in PE [28]. The students emphasise the importance of physical activity and its impact on both physical and mental health. They believe that all pupils, from 1st to 13th grade, should learn about their own bodies and have social interaction experiences. Furthermore, the students state that PE is particularly important for those who are not physically active in their leisure time. PL as a health-promoting activity is the cosmos that can best reflect this theme, where the biggest driving force for developing this form of literacy among students through PL globally is from the increasing degree of inactivity that not only threatens health but also the economy [63].

As students see the sports focus in PE as negative, this, together with the desired focus on mental health, can help turn the subject in a positive direction, as recent research has seen that such a sports focus can have a negative impact on the mental health of those pupils who do not master participation in a sports-dominated PE [29]. Our findings can be seen somewhat in contrast to other research, which has shown that it is students with negative experiences from their own participation as pupils in PE who highlight health discourse as important in the subject [9]. We would still like to point out that the mentioned research was conducted among Spanish students, who themselves have been pupils in a PE subject more characterised by a sports discourse than what our Norwegian students have experienced as pupils. We could not find any contradictions in the student responses in this theme, as pointed out in the aforementioned study. In contrast, many students highlighted the health discourse as important, even though they were also active in sports in their leisure time, and few students highlighted the importance of engaging in activity without placing this in a context of motivation, learning, health, or Bildung.

### 4.1.4. Bildung

The students believe that PE contains elements that have a formative effect on the pupils such as in the development of social skills or elements that can promote self-development. "The positive thing is that the pupils are trained in cooperation and incorporate fair play. Social skills are also trained within PE" (male student). The students are of the opinion that the idea of community and collaboration must be prominent in PE, but also that of working independently. For example, the students mention that PE can help structure the pupils' everyday life, as this subject often has rules that must be followed. Adhering to rules and fellow pupils can help the pupils in everyday life, and these are skills pupils can find useful in the future. Furthermore, the students' perception of PE shows that the Bildung potential in the subject is strong, for example, with respect to thoughts about inclusion. "Through physical education, pupils learn to tolerate and respect that no one is the same, and that you have different prerequisites" (male student). The students believe that inclusion is important in school, that everyone should feel part of a community, and that PE is a good arena for this—also when it comes to the inclusion of new pupils. The same is said about the inclusion of pupils with a multicultural background and who have a language barrier. PL as phenomenological embodiment is the only cosmos within



which we can place this theme. Osnes [41] reflects on the connection between Bildung and PL as where meaning is experienced through action and activity and at the same time in relationship and interaction with others. Furthermore, this connection is seen in the light of Kalfki's theory of categorical Bildung and Merleau-Ponty's phenomenology of perception, which "opens up both a theoretical justification and a practical insight to understand physical literacy as a central part of the bildung process" (p. 129). A pragmatic approach to the concept of PL has also been highlighted as important in later research [40].

Although the students emphasise PE as an important arena for developing personal qualities such as cooperation, tolerance, and empathy, concepts such as creativity were not mentioned. Since LK20 mentions creativity on several occasions, e.g., in connection with dance activities, this is something that PETE must focus on. It was also not stated by the students that effort, practice, and perseverance were highlighted as important in PE, despite the fact that this is considered essential in LK20, and that effort is part of the assessment. Overall, we interpret the students' perceptions of PE as a good arena for fulfilling the mandate of the subject regarding Bildung as described in LK20 and in the values and principles for primary and secondary education.

## 5. Conclusions

The students perceive PE as an important subject in many ways, and expressed this through four themes that we created through the analysis of their responses to the question "How do you perceive physical education?". The students expressed as important the fact that the pupils in PE obtain a lot of physical activity through a variety of activities. Furthermore, the students believe that physical learning and motivation are connected and are mutually important for each other, and that the joy of movement is central to the subject. We interpret this to mean that health discourse is strong in the students' perception of the subject, and that PE is an important contributor to the pupils' Bildung. As we interpret the students' perception of PE, their perception is well reflected in the cosmos of PL as phenomenological embodiment. This is also the cosmos that we recognise in all the themes that were created in the analysis. Based on monistic principles, the Whitehead [64] description of PL as "the motivation, confidence, physical competence, knowledge and understanding to value and take responsibility for engagement in physical activities for life" (p. 83) will provide a good picture of the students' perception of the subject. Although we see that the students who want to teach the subject are largely involved in traditional (sport) activities in their leisure time, we see that they also have many opinions about the strengths and weaknesses of the practice of the subject today. Through how the students perceive the subject, expressed through the four discussed themes, we argue that they have a view of the subject that is largely in line with current governing documents in PE (LK20) and the concept of PL. These findings are somewhat at odds with previous research. We do not find that the students are concerned with concepts such as effort, practice, outdoor life, dancing, swimming, and lifesaving, even though these are topics that are specifically mentioned in LK20. We cannot say with certainty that the students think these parts of PE are unimportant just on the basis that they were not mentioned to any great extent, but with support from studies by González-Calvo and Gerdin [9] and Zoglowek [6], we argue that an important implication of this study is that the students' perceptions and experiences must be considered to be a starting point for change, learning, and development in PETE.

This study must be viewed in the light of some limitations. We had to consider the fact that students in, for example, the first and fourth year of study may have had somewhat different prerequisites for answering the question about their perception of the subject, since during their studies, even through subjects other than PE, they continuously reflect on school subjects and learning. Thus, the students may have developed and changed their perception of PE even before the subject was chosen as part of the teacher training. The fact that all the students who participated belong to the same educational institution must also be considered when interpreting our findings and conclusions.

Furthermore, it will be interesting to see how education, and further working life, will influence the perceptions the students have of the subject, and further, how this is reflected in pedagogical practices.

**Author Contributions:** Conceptualization, O.Ø. and G.O.K.; methodology, O.Ø. and G.O.K.; software, O.Ø.; validation, O.Ø. and G.O.K.; formal analysis, O.Ø. and G.O.K.; investigation, O.Ø. and G.O.K.; resources, O.Ø. and G.O.K.; data curation, O.Ø.; writing—original draft preparation, O.Ø. and G.O.K.; writing—review and editing, O.Ø. and G.O.K.; visualization, O.Ø. and G.O.K.; supervision, O.Ø. and G.O.K.; project administration, O.Ø. and G.O.K. All authors have read and agreed to the published version of the manuscript.

**Funding:** This research received no external funding.

**Institutional Review Board Statement:** The study was conducted in accordance with the Declaration of Helsinki and approved by the Institutional Review Board of Norwegian Agency for Shared Services in Education and Research (NSD) (protocol code #320518 and 15 January 2020).

**Informed Consent Statement:** Informed consent was obtained from all subjects involved in the study.

**Data Availability Statement:** All data generated in the project can be obtained by contacting the corresponding author.

**Acknowledgments:** We thank all participants who responded to the survey in this project.

**Conflicts of Interest:** The authors declare no conflict of interest.

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
