# Peer review of "Norwegian Physical Education Teacher Education Students’ Perceptions of the Subject Physical Education: A Qualitative Study of Students’ Reflections before Starting Their Studies"

_education, doi:10.3390/educsci13050499_

Round 1

Reviewer 1 Report

I found this study to be a well written study. However, there are some minor changes that has to be made. The selection process of the participant need to be calified, also the response rate. Furthermore, asking the question "How do you perceive physical education" with up to 500 words could be a challenge according to short answers from the students. Was this not the case? More information about this is needed. Furthermore, I have some problems with the sentence: "physical learning and motivation are connected and are mutually important for each other, and that the joy of movement is central to the subject. We interpret this to mean that health discourse is strong in the students' perception of the subject". I do not see this connection! This should be clarified.

Author Response

Reviewer 1

Comment

Response

The selection process of the participant need to be calified, also the response rate.

The response is now reported, and the selection process is more clearly described on page 4, subsection Participants in the Methods section.

Furthermore, asking the question "How do you perceive physical education" with up to 500 words could be a challenge according to short answers from the students. Was this not the case? More information about this is needed.

There is an argument (and citation) for this type of survey question included in the subsection Survey in the Methods section now.  

Furthermore, I have some problems with the sentence: "physical learning and motivation are connected and are mutually important for each other, and that the joy of movement is central to the subject. We interpret this to mean that health discourse is strong in the students' perception of the subject". I do not see this connection! This should be clarified.

Our use of “this” relates back to the whole argumentation: “The students expressed as important that the pupils in PE obtain a lot of physical activity through a variety of activities. Furthermore, the students believe that physical learning and motivation are connected and are mutually important for each other, and that the joy of movement is central to the subject.”

This argument is within the health discourse in this context, as we interpreted the student responses.

Reviewer 2 Report

The purpose of this study was to gain insight into how future students in physical education teacher education (PETE) perceive the subject.  I believe that this is an innovative and highly relevant study, but that it requires a number of profound changes to the whole proposal.

Regarding the title, this could be made more specific by adding to it where it has been reazlised. For example: University of...

Physical education teacher students' perceptions of the subject physical 2 education. A qualitative study of students' reflections before starting 3 their studies: University of...

The article should modify the abstract according to the following structure:

- Introduction: the problem statement, objectives and scope of the research are presented. In the abstract, the "state of the art" related to the work should be summarised in one or two sentences. It is also advisable to introduce the objective or hypotheses proposed.

- Methodology: you should explain the methodological design of the research. In this case, limit yourself to stating the type of study, population, main measurements or data sources. Avoid including formulas or equations.

- Results: include the findings of the work. Only the most relevant ones should be mentioned in this part.

- Conclusions: incorporate some implications of the research results and present possible solutions. To end the summary, one or two lines can be dedicated to answering the question "What have we contributed?

Regarding the selection of participants, the total population and the procedure followed for their selection or participation could be described. In order to know whether it is representative or not. This should be justified from the specialised literature.

I consider that the article provides very relevant data for society, and specifically for the area of training in the area of Physical Education, but I consider that the authors should choose to present the results obtained in a different way, as this, together with the specific methodology followed for this study, is not clear. It is a qualitative study, but nowhere is this indicated. The possible research questions which, together with the objectives and hypotheses, would help to better understand the study, are not presented. Furthermore, the instrument used is not clearly presented, nor is it indicated whether the instrument is validated or not. Conclusions to have a strong relevance should be presented based on the objectives and hypotheses. This has not been done by the authors.

I consider that the authors should reorganise the presented study and resubmit it.

Author Response

Reviewer 2

Comment

Response

Regarding the title, this could be made more specific by adding to it where it has been reazlised. For example: University of... Physical education teacher students' perceptions of the subject physical 2 education. A qualitative study of students' reflections before starting 3 their studies: University of...

The title is now edited to include where the study was conducted.

The article should modify the abstract according to the following structure:
- Introduction: the problem statement, objectives and scope of the research are presented. In the abstract, the "state of the art" related to the work should be summarised in one or two sentences. It is also advisable to introduce the objective or hypotheses proposed.
- Methodology: you should explain the methodological design of the research. In this case, limit yourself to stating the type of study, population, main measurements or data sources. Avoid including formulas or equations.
- Results: include the findings of the work. Only the most relevant ones should be mentioned in this part.
- Conclusions: incorporate some implications of the research results and present possible solutions. To end the summary, one or two lines can be dedicated to answering the question "What have we contributed?

The abstract is now revised, and we have checked that it is in line with journal guidelines.

Regarding the selection of participants, the total population and the procedure followed for their selection or participation could be described. In order to know whether it is representative or not. This should be justified from the specialised literature.

There is now some more information on the population and selection (including a citation) in the subsection Participants in the Methods section.

I consider that the article provides very relevant data for society, and specifically for the area of training in the area of Physical Education, but I consider that the authors should choose to present the results obtained in a different way, as this, together with the specific methodology followed for this study, is not clear.

As this is a qualitative study, not understood by a theory, rather connected to a conceptual theoretical framework, we have followed suggestions (by e.g., Braun, V., & Clarke, V. (2022). Thematic analysis. A practical guide. Sage.) and presented findings and discussion within one chapter.

It is a qualitative study, but nowhere is this indicated.

This is indicated in the study title, but this is also now expressed more clearly in the methods section, first paragraph in the Participants subsection.

The possible research questions which, together with the objectives and hypotheses, would help to better understand the study, are not presented.

The research question in now more clearly presented on page 4, last paragraph.

Furthermore, the instrument used is not clearly presented, nor is it indicated whether the instrument is validated or not.

The open-ended question used is now elaborated on in the Survey subsection in the Methods section.

Conclusions to have a strong relevance should be presented based on the objectives and hypotheses. This has not been done by the authors.

We have done some adjustments to the Conclusion section now, highlighting implications.

I consider that the authors should reorganise the presented study and resubmit it.

We have done several changes to the manuscript, based on the comments of all 3 reviewers, and hope these changes meets the expectations and standards of the journal, editors and reviewers.

Reviewer 3 Report

I would like to thank you for the opportunity to review this article and I would like to congratulate the authors for this work. For me, as a physical educator, this topic is very important and has a lot of value. Below are my suggestions and at the end my consideration.

This manuscript investigated student physical education teachers' perceptions of the subject of physical education.

Title: The title is approximate to the problem investigated in the manuscript. As a suggestion, the title should provide more concrete, representative and indicative information about where the research was conducted and the concept of physical literacy, which is really what is being assessed.

Abstract: The abstract does not meet the criteria set out in the journal. The background section is missing. Please adhere to the MDPI guidelines, it is very important for publication. This is the most important section of the document as it will be read many more times than even the manuscript itself, so it needs the utmost attention. A brief note on the importance of the research is an excellent end to a high standard abstract.

Introduction

As mentioned, I find this research extremely important in contributing to the field of Physical Education and Physical Literacy. I agree with the authors' justifications and read many very good and current arguments.  However, it is suggested to provide more information about the importance of physical literacy in this population.

It is suggested to the authors that based on the stated objective they highlight research questions that help to conduct the research and discussion based on the findings in which the study variables, the study population, and the expected outcome appear.

Material and method.

Instruments: It is suggested to include adhere to the journal guidelines. This section is missing. It is very important because it describes the instrument used to assess physical literacy.

Participants. This section should be better defined. The characteristics of the sample should be included. Inclusion-exclusion criteria are not reported....

Statistical analysis. This section should deal with the statistical analysis. I consider that what is stated should be included in a section called "procedure". Please adhere to the MDPI templates for writing manuscripts.

Results:

The findings are displayed correctly and are easy to read and straightforward for any academic.

Discussion: I think you have done a great job in comparing your findings with other studies, but it is too short . It is suggested to include a section on practical and theoretical implications to evaluate the scope of the research.

Conclusions: They are too long and only retell the findings. It is important to highlight the practical implications of these findings for the educational community.

Author Response

 Reviewer 3

Comment

Response

Title: The title is approximate to the problem investigated in the manuscript. As a suggestion, the title should provide more concrete, representative and indicative information about where the research was conducted and the concept of physical literacy, which is really what is being assessed.

The title is now edited to include where the study was conducted.

Abstract: The abstract does not meet the criteria set out in the journal. The background section is missing. Please adhere to the MDPI guidelines, it is very important for publication. This is the most important section of the document as it will be read many more times than even the manuscript itself, so it needs the utmost attention. A brief note on the importance of the research is an excellent end to a high standard abstract.

The abstract is now revised, and we have checked that it is in line with journal guidelines.

Introduction
As mentioned, I find this research extremely important in contributing to the field of Physical Education and Physical Literacy. I agree with the authors' justifications and read many very good and current arguments.  However, it is suggested to provide more information about the importance of physical literacy in this population.

There is now added information of the importance of PL in youth. Page 2, second paragraph.

It is suggested to the authors that based on the stated objective they highlight research questions that help to conduct the research and discussion based on the findings in which the study variables, the study population, and the expected outcome appear.

The research question in now more clearly presented on page 4, last paragraph.

Material and method.
Instruments
: It is suggested to include adhere to the journal guidelines. This section is missing. It is very important because it describes the instrument used to assess physical literacy.

This study did not assess PL, so no instrument was used for that purpose. The open-ended question was the survey used to find answer to the research question in this qualitative study.

Participants. This section should be better defined. The characteristics of the sample should be included. Inclusion-exclusion criteria are not reported....

As this was a natural selection due to the nature of school classes, and the desire to understand those students’ perception, all students were invited. There is now a clarification about this “purposeful sampling”, including the response rate, in the Participants subsection in the Methods section.

Statistical analysis. This section should deal with the statistical analysis. I consider that what is stated should be included in a section called "procedure". Please adhere to the MDPI templates for writing manuscripts.

As this is a qualitative study, there was no statistical analysis beside descriptives of gender balance etc.  

Results: The findings are displayed correctly and are easy to read and straightforward for any academic.

Thank you for the comment.

Discussion: I think you have done a great job in comparing your findings with other studies, but it is too short. It is suggested to include a section on practical and theoretical implications to evaluate the scope of the research.

We have now highlighted the implications in the Conclusion section

Conclusions: They are too long and only retell the findings. It is important to highlight the practical implications of these findings for the educational community.

We have now highlighted the implications in the Conclusion section

Round 2

Reviewer 2 Report

First of all, I thank the authors for making improvements to the manuscript. The quality of the manuscript has improved, but many of the issues raised previously remain unclear, and perhaps now the improvements have brought them to the surface and made them more apparent. There is no concordance with the objective proposed in the abstract: 

"The purpose 7 of this study was to gain insight into how future students in physical education teacher education 8 (PETE) perceive the subject. Written responses from 112 students at the start of their PETE study 9 were analysed within the framework of reflexive thematic analysi....".

Subsequently, in the methodology section, there is only a description of some socio-demographic questions and some other questions referring to the habit of physical education exercise. The instrument used is still not clearly described, which, being a qualitative study, is supposed to be an interview or an open-ended questionnaire. This request has already been made to the authors in previous reviews, and is still missing. The instrument and its review and validation procedure should be described in detail, otherwise the manuscript lacks quality and could never be replicated by other authors.

Furthermore, the procedure of the qualitative data is not described, was a category analysis carried out? What were they? Were any tools used, etc.? Without providing all this information, the methodology of the study is not complete.

The authors were also asked to further specify the objectives of the study by providing hypotheses. In this way, the results would be presented on the basis of these and everything would be more closely related, thus improving the study. If this is done, the conclusions would be presented on the basis of them, i.e. either on the basis of the objectives and confirming or refuting hypotheses, or on the basis of the hypotheses directly.

I believe that many improvements have been made to the study but it does not yet have the methodological rigour to be considered for publication in the journal. Authors are encouraged to make improvements and resubmit the article again

Author Response

Reviewer 2

Comment

Response

First of all, I thank the authors for making improvements to the manuscript. The quality of the manuscript has improved, but many of the issues raised previously remain unclear, and perhaps now the improvements have brought them to the surface and made them more apparent. There is no concordance with the objective proposed in the abstract: 

"The purpose 7 of this study was to gain insight into how future students in physical education teacher education 8 (PETE) perceive the subject. Written responses from 112 students at the start of their PETE study 9 were analysed within the framework of reflexive thematic analysi....".

We thank your for the comment, and hope to clarify some of the concerns in this response letter.

Subsequently, in the methodology section, there is only a description of some socio-demographic questions and some other questions referring to the habit of physical education exercise.

The instrument used is still not clearly described, which, being a qualitative study, is supposed to be an interview or an open-ended questionnaire. This request has already been made to the authors in previous reviews, and is still missing. The instrument and its review and validation procedure should be described in detail, otherwise the manuscript lacks quality and could never be replicated by other authors.

The main object in this study was to investigate “How do future Norwegian PETE students perceive the school subject PE?” This perception was set out to get an understanding of by having students write an answer to the open-ended question “"How do you perceive physical education in primary school?". This is described in line 175-182. There were no other instruments used in regard to that research question. Other questions were about gender, age activity type, to describe the sample, and not to include in the discussion about the research question. Therefor, there is no instrument to describe deeper nor validate. If we have measured activity level etc, that would have to be included.

The question about replication of a qualitative study brings us into an assumption of lack of deep understanding of qualitative research in general, Biq Q qualitative research and especially reflexive thematic analysis (Big Q qualitative: Research that retain a focus on individual experience, understandings, views, beliefs and motivations, their actions, and behaviours, but recognising this as thoroughly socially and situated within a wider social context. https://www.thematicanalysis.net/lecture-resources/).

Furthermore, the procedure of the qualitative data is not described, was a category analysis carried out? What were they? Were any tools used, etc.? Without providing all this information, the methodology of the study is not complete.

This comment is somewhat hard to understand. The whole sub-section called “Analysis” (Line 195-239) is dedicated to shed light on this procedure.

The authors were also asked to further specify the objectives of the study by providing hypotheses. In this way, the results would be presented on the basis of these and everything would be more closely related, thus improving the study. If this is done, the conclusions would be presented on the basis of them, i.e. either on the basis of the objectives and confirming or refuting hypotheses, or on the basis of the hypotheses directly.

Again, the use of hypothesis is on several layers very against the main assumptions of qualitative research in general, Biq Q qualitative research and especially when using reflexive thematic analysis.

Reviewer 3 Report

Congratulations on the improvements. The manuscript is much improved. Thank you for adhering to our suggestions and I encourage you to adhere to your research and share new findings with the scientific community.

Author Response

 Reviewer 3

Comment

Response

Congratulations on the improvements. The manuscript is much improved. Thank you for adhering to our suggestions and I encourage you to adhere to your research and share new findings with the scientific community.

Thank you for the comment